# Association between Migrant Women’s Legal Status and Prenatal Care Utilization in the PreCARE Cohort

**DOI:** 10.3390/ijerph17197174

**Published:** 2020-09-30

**Authors:** Maxime Eslier, Catherine Deneux-Tharaux, Priscille Sauvegrain, Thomas Schmitz, Dominique Luton, Laurent Mandelbrot, Candice Estellat, Elie Azria

**Affiliations:** 1Université de Paris, CRESS, Obstetrical Perinatal and Pediatric Epidemiology Research Team, EPOPé, INSERM, INRA, 75014 Paris, France; maxime.eslier@hotmail.fr (M.E.); catherine.deneux-tharaux@inserm.fr (C.D.-T.); priscille.sauvegrain@inserm.fr (P.S.); thomas.schmitz@aphp.fr (T.S.); 2Department of Obstetrics and Gynaecology, Robert Debré Hospital, AP-HP, Université de Paris, 75019 Paris, France; 3Department of Obstetrics and Gynaecology, Beaujon-Bichat Hospital, FHU PREMA, AP-HP, Université de Paris, 75018 Paris, France; dominique.luton@aphp.fr; 4Department of Obstetrics and Gynaecology, Louis Mourier Hospital, FHU PREMA, AP-HP, Université de Paris, 92700 Colombes, France; laurent.mandelbrot@aphp.fr; 5AP-HP, Hôpitaux Universitaires Pitié Salpêtrière - Charles Foix, Public Health Department, CIC 1425-EC, Sorbonne Université, INSERM, Institut Pierre Louis d’Epidémiologie et de Santé Publique, Team PEPITES, 75013 Paris, France; candice.estellat@aphp.fr; 6Maternity Unit, Groupe Hospitalier Paris Saint Joseph, FHU PREMA, Université de Paris, 75014 Paris, France

**Keywords:** legal status, undocumented migrant, prenatal care utilization, health inequalities

## Abstract

Barriers to access to prenatal care may partially explain the higher risk of adverse pregnancy outcomes among migrants compared with native-born women in Europe. Our aim was to assess the association between women’s legal status and inadequate prenatal care utilization (PCU) in France, where access to healthcare is supposed to be universal. The study population was extracted from the PreCARE prospective cohort (N = 10,419). The associations between women’s legal status and a composite outcome variable of inadequate PCU were assessed with multivariate logistic regressions. The proportion of women born in sub-Saharan Africa (SSA) was higher among the undocumented than that of other migrants. All groups of migrant women had a higher risk of inadequate PCU (31.6% for legal migrants with European nationalities, 40.3% for other legal migrants, and 52.0% for undocumented migrants) than French-born women (26.4%). The adjusted odds ratio (aOR) for inadequate PCU for undocumented migrants compared with that for French-born women was 2.58 (95% confidence interval 2.16–3.07) overall, and this association was similar for migrant women born in SSA (aOR 2.95, 2.28–3.82) and those born elsewhere (aOR 2.37, 1.89–2.97). Regardless of the maternal place of birth, undocumented migrant status is associated with a higher risk of inadequate PCU.

## 1. Introduction

While Europe has sometimes been a land of refuge, it has for several years been undergoing what has been called a migration crisis and also a humanitarian crisis. Migrants comprise a significant and growing proportion of childbearing women in these countries [1]. According to the French National Institute of Statistics and Economic Studies (INSEE), 23.2% of women who gave birth in France in 2018 were born elsewhere [2].

Migrant women in Europe have a higher risk of adverse maternal and perinatal outcomes than women born in the host country, including a higher risk of maternal mortality [3] and of severe maternal [4,5,6] and perinatal morbidity [7,8,9,10,11]. These risks are higher among migrant women from sub-Saharan Africa (SSA) than among women born in other geographical areas [5,10,12,13,14,15,16].

The mechanisms explaining these social inequalities in health are complex, numerous, and unclear. Adequate prenatal care utilization (PCU), assessed both by its timing and its content, is believed to be an important factor in reducing maternal and perinatal risk [17]. Migrant status has been associated with inadequate PCU in several high-income countries [18,19]. Furthermore, rates of inadequate PCU among migrant women vary widely by their place of birth, with the highest rates seen in women from sub-Saharan Africa (SSA) [18,19]. Inadequate PCU might therefore play a role in the mechanisms of health inequalities between migrant and nonmigrant populations. Because databases rarely collect legal status [20], very little is known about how migrant women’s legal status, especially undocumented status, which affects and can be one of the multiple barriers to PCU.

Because France has chosen the principle of universal access to healthcare [21], particularly during pregnancy and including authorization of free care for undocumented migrants, it offers an opportunity to assess actual access to prenatal care according to migrants’ legal status, in a setting where policies make it theoretically possible for all. The French multicenter prospective PreCARE cohort is thus one of the few to have collected this status from migrant pregnant women.

The aim of our analysis of the PreCARE cohort data was first to assess the association between maternal legal status and inadequate PCU, and second to explore the role of maternal birthplace in this association.

## 2. Materials and Methods

The French PreCARE multicenter prospective cohort study took place in four university hospital maternity units in the northern Paris area from October 2010 to May 2012 [17,19]. This geographical area is characterized by its high prevalence of social deprivation and its multicultural population. The study included all pregnant women ≥18 years old, registered and giving birth in these hospitals. This analysis covers the study population of women who gave birth after 21 completed weeks of gestation. It excluded women who finally gave birth in a nonparticipating hospital, were lost to follow-up, or had completely empty questionnaires.

The regional ethics review board CPP-Ile-de-France III (no. 09.341bis, approved November 19, 2009) and the CNIL (Commission Nationale Informatique et Liberté) approved this study. Each woman provided oral informed consent, in accordance with French law.

We collected social and demographic characteristics (age, deprivation index, schooling level, social welfare coverage at inclusion, maternal birthplace, length of residency, language barrier) and legal status by self-administered questionnaires, once at inclusion and again during the postpartum period before discharge. To enable the inclusion of women not speaking French fluently or who could not read or write, these questionnaires were available in the four principal languages of the main region of origin of the residents, and a research assistant or interpreter helped in their completion when needed. Women’s medical history and information about their pregnancy, labor, delivery, and postpartum period were collected by research assistants and practitioners (midwives and obstetricians) by specific questionnaires completed in the postpartum period before discharge. Research assistants also collected PCU data from the postpartum questionnaire and the women’s medical files.

The exposure of interest, the women’s legal status, was first categorized in four groups: (1) nonmigrants; (2) migrants with French or European nationality (thus automatically legal); (3) other legal migrants, without French or European nationality; and (4) undocumented migrants. Nonmigrants were women born in France. Migrant women with French or European nationality were born outside France and reported a French or other European nationality. Other legal migrants were born outside France and held a residence permit, or a temporary residence permit, or a short- or long-term tourist visa, issued by French or European authorities. Asylum seekers and women with refugee status were categorized as legal migrants. Undocumented migrants were born outside France, had a non-European nationality, and were awaiting a decision about their legal status (regularization). Information about legal status was self-reported during the inclusion questionnaire administered either by the women herself or a research assistant or interpreter when needed. If this information was missing, it was extracted from the postpartum questionnaire.

In a secondary analysis to consider the women’s birthplace together with their legal status and based on the reported increased risk for migrants from SSA [5,10,12,14,15,16,19], the exposure of interest was categorized in seven groups: (1) nonmigrants, (2) migrants with French or European nationality, not born in SSA, (3) migrants with French or European nationality born in SSA, (4) other legal migrants not born in SSA, (5) other legal migrants born in SSA, (6) undocumented migrants born somewhere else besides SSA, and (7) undocumented migrants born in SSA.

The main outcome for the primary and secondary analyses was inadequate PCU, categorized as a binary variable. PCU was assessed with three components: (i) late initiation of prenatal care (>14 weeks of gestation), (ii) proportion of prenatal visits completed of the number recommended according to gestational age at delivery (extra visits to check maternal blood pressure or for fetal heart monitoring were not counted in the number of visits), and (iii) absence of the recommended ultrasound scans in the first (at 11–14 weeks), second (21–24 weeks), and third trimesters (31–34 weeks). These three components were integrated into an index of PCU adapted from the Adequacy of Prenatal Care Utilization index (APNCU) [22,23] for the French prenatal care guidelines. PCU was considered inadequate if care did not begin until 14 completed weeks of gestation, or if the percentage of prenatal visits was <50% of the recommended number, or if the first trimester ultrasound or both of the later ultrasounds were missing. The precise method of calculating this modified APNCU (mAPNCU) has been described in a previous publication [17]. Other items that may be indirect indicators of a lack of prenatal care were also reported.

We characterized maternal social deprivation at the beginning of pregnancy by a quantitative deprivation index that was the sum of four dimensions of deprivation: social isolation, poor or insecure housing conditions, no work-related household income, and no permanent healthcare insurance. This index has been previously described [19].

Baseline characteristics and PCU were first described by the women’s legal status. Qualitative variables were expressed as percentages, quantitative variables by their medians, and interquartile ranges. The statistical tests used were the Kruskal–Wallis test for medians, and the chi-square test (or Fisher’s exact test, as appropriate) for qualitative variables.

Logistic regression models were used to assess the association between legal status and inadequate PCU. Causal assumptions between legal status, inadequate PCU, and covariates were represented with a directed acyclic graph to depict the exposure–outcome relations with confounding and intermediate factors. This graph helped to select variables that are confounders (i.e., variables associated with both the exposure, which is legal status, and the outcome of inadequate PCU, and not on the causal pathway between legal status and inadequate PCU) and those that do not qualify as confounders (especially intermediate factors on the causal pathway) [24]. The main regression model included only confounders: maternal age, education level, and number of previous pregnancies. We also adjusted for the maternity unit of delivery. The linearity of the association of the continuous variables (age and number of previous pregnancies) with legal status was tested. As the association with maternal age was not linear, this variable was categorized. Maternity unit effects were handled as fixed effects.

To exclude the influence of other health systems than that of France, a sensitivity analysis excluded women who had arrived in France less than 12 months before delivery.

The proportion of women with missing data after adjustment in the multivariate model was 7.6%. Multiple imputation using chained equations (25 datasets) was performed to handle the missing data, assumed to be missing at random (MAR) [25]. The results are presented with imputed data as adjusted odds ratios (aOR) with their 95% confidence intervals (95% CI). All statistical tests were two-tailed and the threshold for statistical significance was set at a probability value of <0.05. Analyses were performed with STATA software, version 13.1 (Stata Corporation, College Station, TX, USA).

## 3. Results

Among the 10,576 women asked to participate in the Pre-CARE study, 10,419 agreed (98.5%). After excluding women mistakenly included (*n* = 60), those who withdrew their consent (*n* = 6), those who gave birth before 21 completed weeks of gestation (*n* = 135), or in a nonparticipating maternity unit (*n* = 209), those lost to follow-up (*n* = 378), and those with missing questionnaires (*n* = 32), the analysis included 9599 women (Figure 1).

In the study population, 4523 women were born in France (47.1%), 1555 were migrants with French or European nationality (16.2%), 2806 were other legal migrants (29.2%), and 715 were undocumented migrants (7.4%). Table 1 summarizes the women’s baseline characteristics by legal status. Compared with nonmigrants, migrant women, especially undocumented ones, were more often socially deprived and faced a language barrier. The undocumented migrants had also lived in France for less time than the other migrant groups. The proportion of women born in SSA was higher among undocumented migrants than in the other migrant groups.

Migrant women, especially undocumented ones, had a higher frequency of inadequate PCU according to the mAPNCU index compared with nonmigrants (52.0% vs. 26.4%, *p* < 0.001). The PCU was globally inadequate because care began too late and because the first trimester ultrasound was frequently not performed. The differences observed between groups were greater for early prenatal care, such as the first trimester ultrasound, than for later care, such as the third trimester ultrasound or the pre-anesthesia evaluation (Table 2).

In the multivariate analysis, all the groups of migrant women had a higher risk of inadequate PCU than the nonmigrant women (Table 3). Nonetheless, the strength of the association increased from migrant women with French or European nationality to undocumented migrant women, with the latter at highest risk (aOR 2.58, 95% CI 2.16–3.07).

When migrant women were differentiated by place of birth, associations between legal status and PCU were similar overall, with an increased risk of inadequate PCU in all groups of migrants compared with French-born women, although they did not reach statistical significance in the comparison with migrants of French or other European nationality. The risk of inadequate PCU was again highest for undocumented migrants, regardless of place of birth (Table 4), significantly higher than in the other three groups. When analyzed in the strata of legal non-European migrant and undocumented migrant women, the rate of inadequate PCU was higher for women born in SSA than for those born elsewhere (Appendix A
Table A1).

The sensitivity analysis with complete cases showed similar results (Appendix A
Table A2), as did that excluding women who arrived in France less than 12 months before delivery (Appendix A
Table A3).

## 4. Discussion

In France, migrant women, compared with those born in France, have a higher risk of inadequate prenatal care utilization. Moreover, our results show a gradient that might be described as social among the migrant women according to legal status, from an adjusted odds ratio of 1.2 for the migrant women with a French or other European nationality to an adjusted odds ratio of 2.6 for the undocumented women, the subgroup at highest risk. Our results also show that these associations exist both for women born in sub-Saharan Africa, as well as for those born elsewhere and suggest that migrant women’s legal status serves as a greater barrier to access to care than their geographical origin itself.

This study is one of the very few studies to tackle the issue of the possible impact of migrant’s legal status on PCU. Its design, based on prospective multicenter data, makes it able to illuminate the association between legal status and PCU for categories beyond accepted asylum seekers and refugees. Most databases do not collect information about legal status. However, legal status can sometimes be obtained by interlinking with administrative databases, as done by Korinek and Smith in 2011 [26].

In our analysis, we chose to examine this status in four groups to understand more clearly the specific impact of lacking documents. Similarly, we chose to isolate women born in SSA from those born elsewhere because previous reports have shown the higher risks of inadequate PCU and morbidity in this subgroup compared with that in the others [5,10,12,14,15,16,19]. The large sample of migrant women, and in particular undocumented migrant women, provides good statistical power. The data collection method, especially the availability of the questionnaires in four different languages and the availability of a research assistant or interpreter to complete it enabled us to include women who did not speak French and reduced both the risk of selection bias and the missing data rate. The high prevalence of social deprivation and the multicultural cohort recruited in this area is, in this context, a strength, even though it produces a population not representative of that of France. The choice to build this cohort in this setting was deliberate and consistent with our scientific objectives, in particular to be able to constitute a large group of migrant and undocumented women so that we could specifically analyze these subgroups with appropriate statistical power. Nevertheless, the substantial number of women excluded for missing data for pregnancy outcomes or with questionnaires, because they delivered elsewhere or were lost to follow-up, remains a limitation. Because these women were more often underprivileged and born abroad than the final sample, we hypothesize that if there is a differential bias, these exclusions may have resulted in underestimating the strength of the association (Appendix A
Table A4). The rate of missing data in the study population was low and, as demonstrated by the comparisons of results obtained by the analyses with imputed data and with complete cases, had a very limited impact on the results. Information about the legal status of these migrant women was self-reported. Although the prevalence of undocumented women is relatively high, we cannot rule out the possibility that it was underestimated.

Our analysis shows that although all the groups of migrant women had a higher risk of inadequate PCU than the nonmigrants, a gradient, essentially social, exists according to their legal status, with undocumented women being the category at highest risk. Interestingly, the highest proportion of undocumented migrants in our cohort were among the women born in SSA, which let us explore the respective contributions of geographical origin and legal status to these barriers to prenatal care. In a previous analysis, we showed that PCU varied according to the women’s region of birth, with the frequency of inadequate PCU highest among women born in SSA [19]. The results of the analysis presented here show that this higher risk of inadequate PCU among migrants born in SSA may actually reflect the higher proportion of undocumented migrants within this subgroup. Categorization of migrants by legal status rather than by region of origin may be more relevant in exploring the mechanisms of barriers to care.

Besides geographical origin, legal status appears to also be an important determinant of inadequate PCU. Only very few studies have investigated PCU specifically in undocumented populations. Wolff et al. described PCU among a group of 134 undocumented migrant women who attended a free antenatal facility in Geneva [27]. Korinek and Smith in 2011 showed from the Utah state administrative database that the legal status of migrant women was one of the important factors influencing prenatal care utilization, with undocumented women being the most at risk of not receiving adequate care [26]. A few studies have reported inadequate PCU with late initiation of care and fewer prenatal visits [28], as well as disparities to accessing prenatal care for women who were, compared with those who were not, refugees [29] or asylum seekers [30]. In our analysis, asylum seekers and women with refugee status were categorized as legal migrants, for unlike undocumented migrants, they have a permit to live in France. Although undocumented migrants in France can claim free care under the state medical assistance (AME) system (created in 1999), their position is precarious. Women can only apply for AME after 3 months in France (demonstrated with evidence), and it takes another two months to be valid and usable. Moreover, some healthcare professionals refuse to accept patients covered by it [31]. In our study, 23.2% of undocumented migrants live without AME, even though more than 90% of them had been in France for more than three months (Table 1). In addition, our sensitivity analysis excluding women who arrived in France less than 12 months before delivery showed a persistently higher risk of inadequate PCU for undocumented migrants compared with nonmigrants (Appendix A
Table A3). This discussion on the French AME joins the discussion on the extension of Medicaid to undocumented pregnant women in the United States, which has been showed to be associated with improved PCU [32] and stresses the importance of being able to activate it very quickly in case of pregnancy.

Asylum seekers and refugees have a status that provides them with some social protection, while lack of documents may be a factor that impairs their ability to interact with the health system. This status is an additional factor to those already affecting migrants in general: language and cultural barriers, social isolation, poverty, discrimination, maternal stress, lack of health literacy, and social protection [9,33,34,35,36,37]. In addition, the permanent risk of arrest and the impossibility of working legally are also factors that can impede access to care despite a system supposed to guarantee it. It is likely that all these factors limit access to care and are involved in causal pathways directly or indirectly between migrant status and increased morbidity [17,18,19,38,39].

## 5. Conclusions

A better understanding of the mechanisms and in particular the role played by national integration policies in this association between legal status and PCU could benefit from international comparisons, in particular among countries with a wide range of Migrant Integration Policy Index ratings (MIPEX) [40]. Regardless of where outside of France a woman was born, as an undocumented migrant she is at higher risk of inadequate prenatal care utilization, despite the French system that is supposed to guarantee universal access to care. This increased risk of inadequate PCU may be associated with higher maternal and perinatal morbidity.

Quicker and easier implementation of rights, especially AME could facilitate access to prenatal care. In addition, urgent testing and implementation of targeted interventions, such as educational programs to strengthen health literacy to help people navigate a complex healthcare system and interventions to make this system more user-friendly, in order to improve access to prenatal care for this growing subgroup in Western countries is essential.

## Figures and Tables

**Figure 1 ijerph-17-07174-f001:**
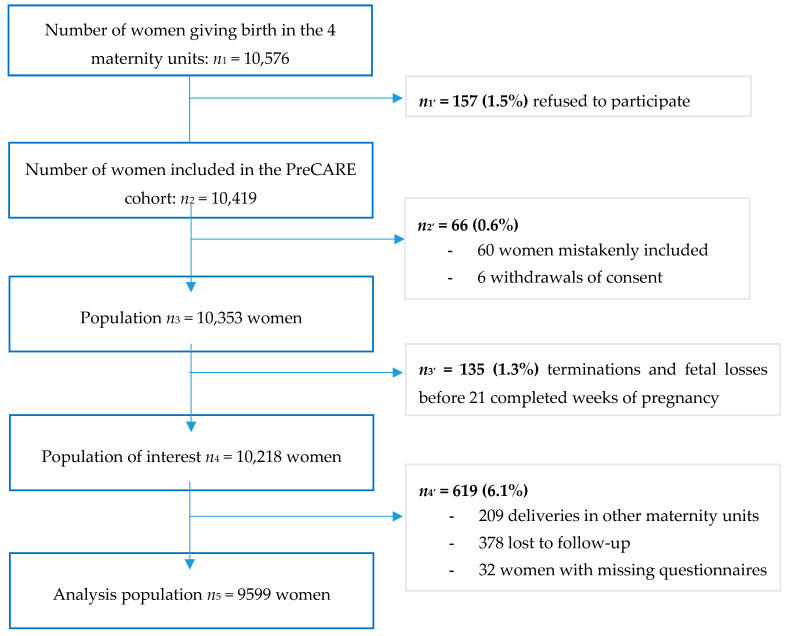
Study sample selection.

**Table 1 ijerph-17-07174-t001:** Baseline characteristics of women according to legal status (*n* = 9599).

	Nonmigrants (*n* = 4523)		Migrants with French or European Nationality (*n* = 1555)		Other Legal Migrants (*n* = 2806)		Undocumented Migrants (*n* = 715)		*p* ****
	*n*	%	*n*	%	*n*	%	*n*	%	
**Age (years) (*n* = 9599)**									<0.001
<20	76	1.7	13	0.8	26	0.9	13	1.8	
[20,21,22,23,24,25]	606	13.4	188	12.1	375	13.4	134	18.7	
[25,26,27,28,29,30]	1428	31.6	362	23.3	871	31.0	251	35.1	
[30,31,32,33,34,35,36,37,38,39,40]	2255	49.9	864	55.6	1379	49.1	295	41.3	
≥40	158	3.5	128	8.2	155	5.5	22	3.1	
Social isolation (*n* = 9596)	77	1.7	55	3.5	172	6.1	123	17.2	<0.001
Poor or insecure housing conditions (*n* = 9596)	412	9.1	175	11.3	510	18.2	406	56.8	<0.001
No standard healthcare insurance (*n* = 9585)	396	8.8	276	17.7	833	29.7	639	89.4	<0.001
No work-related household income (*n* = 9593)	435	9.6	194	12.5	520	18.5	335	46.9	<0.001
**Deprivation index * (*n* = 9502)**									<0.001
0 criterion	3661	80.9	1087	69.9	1574	56.1	0	0.0	
1 criterion	515	11.4	293	18.8	667	23.8	206	28.8	
2 criteria	240	5.3	117	7.5	337	12.0	145	20.3	
3 or 4 criteria	103	2.3	55	3.5	216	7.7	286	40.0	
**Schooling level (*n* = 9504)**									<0.001
≤Primary school	32	0.7	88	5.7	391	13.9	156	21.8	
Middle school	643	14.2	297	19.1	633	22.6	173	24.2	
High school	892	19.7	419	26.9	793	28.3	195	27.3	
University	2943	65.1	736	47.3	943	33.6	170	23.8	
**Social welfare coverage at inclusion (*n* = 9513)**									<0.001
Standard health insurance (SHI)	429	9.5	264	17.0	675	24.1	0	0.0	
SHI + Complementary health insurance	3695	81.7	1013	65.1	1293	46.1	0	0.0	
Universal health coverage (CMU)	366	8.1	218	14.0	607	21.6	0	0.0	
State medical assistance (AME)	1	0.0	22	1.4	90	3.2	473	66.2	
No healthcare insurance	29	0.6	36	2.3	136	4.8	166	23.2	
**Maternal birthplace (*n* = 9583)**									<0.001
Metropolitan France	4357	96.3	0	0.0	0	0.0	6	0.8	
Overseas French territory	166	3.7	0	0.0	0	0.0	0	0.0	
Europe (other)	0	0.0	339	21.8	77	2.7	51	7.1	
North Africa	0	0.0	611	39.3	1314	46.8	191	26.7	
Sub-Saharan Africa	0	0.0	381	24.5	877	31.3	308	43.1	
Asia–Middle East	0	0.0	128	8.8	379	13.6	109	15.2	
Other	0	0.0	86	5.5	159	5.7	50	7.0	
Length of residency (median in months) ** (*n* = 8687) IQR 25/75			141.4 82.7	250.6	71.9 26.0	118.8	31.6 9.4	73.4	<0.001
Language barrier (*n* = 9322)	41	0.9	150	9.6	575	20.5	241	33.7	<0.001
Smoked before pregnancy (*n* = 9507)	1237	27.3	206	13.2	138	4.9	45	6.3	<0.001
Smoked during pregnancy (*n* = 9455)	644	14.2	113	7.3	66	2.4	21	2.9	<0.001
Alcohol during pregnancy (*n* = 9471)	372	8.2	96	6.2	134	4.8	61	8.5	<0.001
Drugs during pregnancy (*n* = 9594)	39	0.9	5	0.3	6	0.2	1	0.1	<0.001
**Prepregnancy BMI (kg/m^2^) (*n* = 9599)**									<0.001
<18.5	303	6.7	76	4.9	122	4.3	46	6.4	
18.5–24.9	2844	62.9	864	55.6	1361	48.5	318	44.5	
25–29.9	767	17.0	366	23.5	733	26.1	167	23.4	
≥30	505	11.2	187	12.0	376	13.4	80	11.2	
Nulliparous (*n* = 9587) High risk at the beginning of pregnancy *** (*n* = 9550)	2302 925	50.9 20.5	520 297	33.4 19.1	963 521	34.3 18.6	332 124	46.4 17.3	<0.001 0.1

IQR, interquartile range; BMI, body mass index; the sum is not equal to 100% due to missing data. * Deprivation index: simple sum of 4 deprivation dimensions: social isolation, poor or insecure housing conditions, no work-related household income, and no permanent heath care insurance. ** If born abroad. *** High-risk pregnancy is defined by at least one of the following items in accordance with French guidelines: history of cardiac disease, hypertension, diabetes, venous thrombosis, pulmonary embolism, Graves’ disease, asthma, homozygous sickle cell, anemia, thrombocytopenia, coagulation disorder, a rare or systemic disease, nephropathy, HIV infection, late miscarriage, preeclampsia, growth restriction, preterm delivery, fetal death, or neonatal death. **** Chi^2^ test (or Fisher’s exact test if necessary) for qualitative variables and Kruskal–Wallis test for medians of quantitative variables.

**Table 2 ijerph-17-07174-t002:** Prenatal care utilization, according to the woman’s legal status (*n* = 9599).

	Nonmigrants (*n* = 4523)		Migrants with French or European Nationality (*n* = 1555)		Other Legal Migrants (*n* = 2806)		Undocumented Migrants (*n* = 715)		*p* ***
	*n*	%	*n*	%	*n*	%	*n*	%	
Initiation of care ≥14 GW (*n* = 9586)	641	14.2	238	15.3	524	18.7	214	29.9	<0.001
Percentage of recommended prenatal visits <50% * (*n* = 9566)	118	2.6	35	2.3	85	3.0	43	6.0	<0.001
First trimester ultrasound not performed between 11 and 14 GW (*n* = 9256)	781	17.3	349	22.4	857	30.5	304	42.5	<0.001
Second trimester ultrasound not performed between 21 and 24 GW (*n* = 9391)	689	15.2	312	20.1	606	21.6	204	28.5	<0.001
Third trimester ultrasound not performed between 31 and 34 GW (*n* = 9433)	753	16.6	278	17.9	584	20.8	160	22.4	<0.001
**Inadequate prenatal care according to mAPNCU index ****	**1196**	**26.4**	**491**	**31.6**	**1131**	**40.3**	**372**	**52.0**	**<0.001**
Missing data for an item of the index	307	6.8	121	7.8	171	6.1	43	6.0	
**Indirect indicators of prenatal care**									
Pre-anesthesia evaluation ≥37 GW (*n* = 9061)	438	9.7	160	10.3	347	12.4	91	12.7	0.001
No determination of Rhesus group before entering the delivery room (*n* = 9531)	23	0.5	7	0.5	15	0.5	4	0.6	0.9
No hepatitis B serology determination before entering the delivery room (*n* = 9550)	33	0.7	6	0.4	15	0.5	3	0.4	0.4
No predelivery identification of estimated fetal weight >95th percentile (among 479 women with birth weight >95th percentile, *n* = 9506)	71	39.0	32	32.7	65	38.0	13	46.4	0.6
No predelivery identification of estimated fetal weight <3rd percentile (among 343 women with birth weight <3rd percentile, *n* = 9439)	58	34.9	25	44.6	41	50.0	8	20.5	0.007
No predelivery identification of placenta previa (among 124 women with placenta previa, *n* = 9422)	5	9.1	3	15.0	3	8.6	0	0	0.5
No predelivery identification of multiple pregnancy (among 141 women with multiple pregnancy, *n* = 9547)	13	19.4	0	0	6	14.3	1	12.5	0.1
No predelivery identification of uterine scar (among 1311 women with a uterine scar, *n* = 9520)	27	5.7	14	5.6	25	5.3	8	7.2	0.9
No predelivery identification of breech presentation (among 392 women with breech delivery, *n* = 9511)	22	10.9	5	9.4	11	10.4	2	6.5	0.9

GW, gestation weeks; * Percentage of recommended prenatal visits according to pregnancy duration. ** mAPNCU (modified the Adequacy of Prenatal Care Utilization) index, which considers initiation of care, percentage of recommended prenatal visits made, and ultrasound scans performed. *** Chi^2^ test (or Fisher’s exact test if necessary) for qualitative variables.

**Table 3 ijerph-17-07174-t003:** Association between legal status and inadequate prenatal care.

	Inadequate Prenatal Care *	
	OR [95% CI]	aOR [95% CI] ^1^
Nonmigrants (*n* = 4523)	1	1
Migrants with French or European nationality (*n* = 1555)	1.31 [1.16–1.49]	1.17 [1.03–1.33]
Legal migrants (*n* = 2806)	1.90 [1.71–2.10]	1.60 [1.43–1.78]
Undocumented migrants (*n* = 715)	3.13 [2.65–3.70]	2.58 [2.16–3.07]

OR, odds ratio; aOR, adjusted odds ratio; CI, confidence interval. * Based on the mAPNCU (modified Adequacy of Prenatal Care Utilization index), which considers initiation of care, percentage of recommended prenatal visits made, and ultrasound scans performed. ^1^ Adjusted for maternal age (dummy variable with 5 classes), maternity unit of delivery, education level, number of previous pregnancies.

**Table 4 ijerph-17-07174-t004:** Association between legal status/maternal birthplace and inadequate prenatal care.

	Inadequate Prenatal Care *	
	OR [95% CI]	aOR [95% CI] ^1^
Nonmigrants (*n* = 4523)	1	1
Migrants with French or European nationality not born in SSA (*n* = 1174)	1.25 [1.09–1.44]	1.15 [0.99–1.34]
Migrants with French or European nationality born in SSA (*n* = 381)	1.52 [1.21–1.91]	1.22 [0.97–1.55]
Other legal migrants not born in SSA (*n* = 1929)	1.73 [1.54–1.94]	1.54 [1.37–1.75]
Other legal migrants born in SSA (*n* = 877)	2.35 [2.02– 2.73]	1.80 [1.52–2.12]
Undocumented migrants not born in SSA (*n* = 407)	2.75 [2.22–3.40]	2.37 [1.89–2.97]
Undocumented migrants born in SSA (*n* = 308)	3.72 [2.92–4.75]	2.95 [2.28–3.82]

OR, odds ratio; aOR, adjusted odds ratio; CI, confidence interval; SSA, sub-Saharan Africa. * Based on the mAPNCU (modified Adequacy of Prenatal Care Utilization index), which considers initiation of care, percentage of recommended antenatal visits made, and ultrasound scans performed. ^1^ Adjusted for maternal age (dummy variable with 5 classes), maternity unit of delivery, education level, number of previous pregnancies.

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
