# Peer review of "Association between Migrant Women’s Legal Status and Prenatal Care Utilization in the PreCARE Cohort"

_ijerph, 2020, doi:10.3390/ijerph17197174_

Round 1

Reviewer 1 Report

The paper deals with a topic of high scientific interest, and the approach used, based on the relationship between the legal status of immigrants in Europe and the inadequate PCU, is novel.

The breadth of the sample and the explanation of its selection, as well as the method used, make the paper a manuscript of enormous quality.

For its part, the format of tables, bibliography and presentation of data, discussion and conclusions are in accordance with the requirements of the journal.

Nevertheless, I consider that there are some modifications that should be made for its publication:

Although in the introduction it is argued that the object of study is a complex phenomenon with a multitude of factors, the legal status is taken as the reference factor before an inadequate PCU. However, other factors that have been the subject of very forceful scientific publications are not mentioned, such as ethnic differences (see Chote, A. A.; Koopmans, G. T.; Redekop, W. K.; et al. (2011)) and other barriers encountered by the migrant population ( see Colon-Burgos, J. F.; Colon-Jordan, H. M.; Reyes-Ortiz, V. E.; et al. (2014)).

In the discussion it is described that the study is the first to associate legal status and the PCU, however, there are studies that although they do not use the same method, they do reach similar conclusions as in the case of the study by Korinek and Smith (2011) entitled "Prenatal care among immigrant and racial-ethnic minority women in a new immigrant destination: Exploring the impact of immigrant legal status" which studies the immigrant population, especially the population of Hispanic origin, in the USA. For this reason, it is necessary to carry out an in-depth review of the literature published in this area and cite it in the paper to provide greater strength to the discussion and conclusions.

In this same line, it is observed that of the 33 references cited, 16 of them (48.5%) are published prior to 2015, and there are numerous references published prior to 2010. An update of the bibliography is required since the phenomenon of study is current and the conditions of immigrants change notably in a few years.

According to what has been argued, I consider that the article is publishable after a minor revision.

Author Response

We are grateful to you for the questions raised and the suggestions made to improve the quality of our paper.

Although in the introduction it is argued that the object of study is a complex phenomenon with a multitude of factors, the legal status is taken as the reference factor before an inadequate PCU. However, other factors that have been the subject of very forceful scientific publications are not mentioned, such as ethnic differences (see Chote, A. A.; Koopmans, G. T.; Redekop, W. K.; et al. (2011)) and other barriers encountered by the migrant population ( see Colon-Burgos, J. F.; Colon-Jordan, H. M.; Reyes-Ortiz, V. E.; et al. (2014)).

We thank the reviewer 1 for this comment and for the references.

For the sake of brevity, in the introduction we have avoided listing the factors that come into play in the complex mechanisms of these social inequalities in health. However, we have cited most of these factors in the discussion section: “This status is an additional factor to those already affecting migrants in general: language and cultural barriers, social isolation, poverty, discrimination, maternal stress, lack of health literacy, and social protection [9,29,30]. In addition, the permanent risk of arrest and the impossibility of working legally are also factors that can impede access to care despite a system supposed to guarantee it. It is likely that all of these factors limit access to care and are involved in causal pathways directly or indirectly between migrant status and increased morbidity [17–19,31,32].”

The two references proposed by the reviewer 1 seem to us to be very relevant and have been included in our references list. The Dutch study by Choté et al conducted from the Generation R Study to support the multidimesional origin of ethnic inequalities in access to prenatal care and that of Colon-Burgos et al to open to a different context which is that of the migration of women from the Dominican Republic to Puerto Rico.

In the discussion it is described that the study is the first to associate legal status and the PCU, however, there are studies that although they do not use the same method, they do reach similar conclusions as in the case of the study by Korinek and Smith (2011) entitled "Prenatal care among immigrant and racial-ethnic minority women in a new immigrant destination: Exploring the impact of immigrant legal status" which studies the immigrant population, especially the population of Hispanic origin, in the USA. For this reason, it is necessary to carry out an in-depth review of the literature published in this area and cite it in the paper to provide greater strength to the discussion and conclusions.

We agree with the reviewer 1, this study is not the first one to associate legal status and PCU and the discussion section has been modified in accordance with that comment.

The paragraph starting by “To our knowledge, this study is the first one based on prospective multicenter data able to Illuminate the association …” has been modified as follow: “This study is one of the very few studies to tackle the issue of the possible impact of migrant’s legal status on PCU. It’s design based on prospective multicenter data make it able to illuminate the association between legal status and PCU for categories beyond accepted asylum seekers and refugees. However, legal status can sometimes be obtained by interlinking with administrative databases, as did Korinek and Smith in 2011 (Korinek).”

The phrase “Although no previous studies…” has been modified as follow: “Only very few studies have investigated PCU specifically in undocumented populations. Wolff et al. described PCU among a group of 134 undocumented migrant women who attended a free antenatal facility in Geneva (Wolff et al 2005). Korinek and Smith in 2011 showed from the Utah state administrative database that the legal status of migrant women was one of the important factors influencing prenatal care utilization, with undocumented women being most at risk of not receiving adequate care (Korinek and Smith 2011).”

As suggested, several references have been added to provide greater strength to the discussion:

The report of the natural experiment in Nebraska by Atkins et al has been added to the reference list with the following comment in the discussion section : “ This discussion on the French AME joins the discussion on the extension of Medicaid to undocumented pregnant women in the United States, which has been showed to be associated with improved PCU (Atkins 2018) and stresses the importance of being able to activate it very quickly in case of pregnancy.”

Although relatively old, the reference to the Geltman and Myers interesting study has also been added such as the pore recent paper by Wolff et al.

In this same line, it is observed that of the 33 references cited, 16 of them (48.5%) are published prior to 2015, and there are numerous references published prior to 2010. An update of the bibliography is required since the phenomenon of study is current and the conditions of immigrants change notably in a few years.

As suggested, our reference list has been updated and three more recent references has been added: Malebranche et al. 2020, Atkins et al. 2018, Colon-Burgos et al 2014.

Reviewer 2 Report

A well written, clearly structured paper with a clear conclusion.

Appendices should be included in the text or deleted.

I would like to have more in the conclusions section, for example, what target interventions to improve prenatal care are recommended?

The last sentence of the discussion would be better suited to the conclusions section.

Author Response

We are grateful to you for the questions raised and the suggestions made to improve the quality of our paper.

- Appendices should be included in the text or deleted.

The reference the Table A4 has been included in the text.

- I would like to have more in the conclusions section, for example, what target interventions to improve prenatal care are recommended? The last sentence of the discussion would be better suited to the conclusions section.

The conclusion section has been modified as follow:

“A better understanding of the mechanisms and in particular the role played by national integration policies in this association between legal status and PCU could benefit from international comparisons, in particular among countries with a wide range of Migrant Integration Policy Index ratings (MIPEX) [33]. Regardless of where outside of France a woman was born, as an undocumented migrant she is at higher risk of inadequate prenatal care utilization, despite the French system that is supposed to guarantee universal access to care. This increased risk of inadequate PCU may be associated with higher maternal and perinatal morbidity.

Quicker and easier implementation of rights, especially AME could facilitate access to prenatal care. In addition, urgent testing and implementation of targeted interventions, such as educational programs to strengthen health literacy to help people navigating in a complex healthcare system and interventions to make this system more user-friendly, in order to improve access to prenatal care for this growing subgroup in Western countries is essential. “

Reviewer 3 Report

For the authors’ guidance, my evaluation and some constructive remarks that would help to improve the paper’s quality are included below:

(1) First of all, the paper would benefit from some serious revisions. Overall, the paper lacks a clear research argument. Why the authors did not specify particular research inquiries that are relevant to the study? Likewise, adding some follow up questions can be beneficial. In case when research questions are identified, then readers can easily conceive the framework of investigation at first glance. They understand somehow “what” the paper attempts to do and they get the “how”, but the most interesting thing, the “why!!!” cannot be systematically conceived very well. For their next scientific contribution, I recommend the authors to come up with their research questions, and why the research is interesting and relevant to the field.

(2) I believe the authors will be more successful if they focus on clear, direct arguments that build up one at a time and add up sequentially to persuasive and tightly focused cases.  My own struggles with the task of writing scientific articles suggest that it is worth spending a very long time in the detailed planning of papers around a sequence of bullet-point arguments. Plans are best if argument-driven.  It is worth resisting writing full prose until a very detailed plan has been developed that seems to work, and then supporting material can be added after each bullet point argument.  Close liaison with a researcher experienced in writing papers in top journals can also be a great help. Certainly, I believe the authors are investigating interesting research areas where there is good scope for innovative new contributions and publications.

(3) I think the authors have successfully linked up some perspectives and concepts of various disciplines. However, this nexus and interrelationships between different disciplines ought to be reinforced, systematically. I recommend quoting the following reference source: de Haas, H., Miller, M. J., & Castles, S. (2020). The Age of Migration: International Population Movements in the Modern World. Red Globe Press.

(4) The methodology and research paradigm of the paper seems a bit vague. There are little substantial clarifications. The paper needs much more work in relation to its structure, methodology, objectives, and discussion. Moreover, the authors may enhance the methodology of their research by citing the publications of E.G. Guba, Y.S. Lincoln, N.K. Denzin and so forth.

(5) I recommend the authors to clarify their methodology in detail, making sure that their planned methods/research tools are fully detailed. They ought to give attention to justifying their chosen methodology in terms of demonstrating applicability, adjustment, and usefulness in the paper. Likewise, the authors ought to refine the article by considering “scientific writing techniques” (i.e. building an argument; designing research questions; quotation rules, outlining sections and sub-sections, and so forth). Thus, I would encourage the authors to undertake some revisions, corrections, and paraphrasing works that may take some time.

(6) All in all, I recommend the authors to reconsider the approach adopted here; think about the main empirical question they wish to examine; make sure the literature review is a lot more cohesive, and make sure the link between the research question and empirical results is a lot 'tighter' than presented herewith.

(7) MDPI - Int. J. Environ. Res. Public Health is a remarkable peer-reviewed journal. A referee ought to recommend a manuscript that is deemed a great contribution to the journal’s future achievements. Consequently, the paper demonstrates rigorous research outcomes/findings that can be useful for the readers of the MDPI - Int. J. Environ. Res. Public Health.

Author Response

(1) First of all, the paper would benefit from some serious revisions. Overall, the paper lacks a clear research argument. Why the authors did not specify particular research inquiries that are relevant to the study? Likewise, adding some follow up questions can be beneficial. In case when research questions are identified, then readers can easily conceive the framework of investigation at first glance. They understand somehow “what” the paper attempts to do and they get the “how”, but the most interesting thing, the “why!!!” cannot be systematically conceived very well. For their next scientific contribution, I recommend the authors to come up with their research questions, and why the research is interesting and relevant to the field. 

We would like to thank the reviewer for this general advice, which we will certainly take into account in our future scientific contributions. We believe, however, that the "why" of this analysis is rather clear in the introduction. 

 (2) I believe the authors will be more successful if they focus on clear, direct arguments that build up one at a time and add up sequentially to persuasive and tightly focused cases.  My own struggles with the task of writing scientific articles suggest that it is worth spending a very long time in the detailed planning of papers around a sequence of bullet-point arguments. Plans are best if argument-driven.  It is worth resisting writing full prose until a very detailed plan has been developed that seems to work, and then supporting material can be added after each bullet point argument.  Close liaison with a researcher experienced in writing papers in top journals can also be a great help. Certainly, I believe the authors are investigating interesting research areas where there is good scope for innovative new contributions and publications. 

We would like to thank the reviewer for sharing his experience with us. 

(3) I think the authors have successfully linked up some perspectives and concepts of various disciplines. However, this nexus and interrelationships between different disciplines ought to be reinforced, systematically. I recommend quoting the following reference source: de Haas, H., Miller, M. J., & Castles, S. (2020). The Age of Migration: International Population Movements in the Modern World. Red Globe Press. 

Thank you for this comment and for the reference of this recently released book that we haven't read yet and which has just been ordered. 

(4) The methodology and research paradigm of the paper seems a bit vague. There are little substantial clarifications. The paper needs much more work in relation to its structure, methodology, objectives, and discussion. Moreover, the authors may enhance the methodology of their research by citing the publications of E.G. Guba, Y.S. Lincoln, N.K. Denzin and so forth. 

We are not sure we understand this comment as the reference cited refers to qualitative research competing paradigms. Our research does not fall within this framework. 

(5) I recommend the authors to clarify their methodology in detail, making sure that their planned methods/research tools are fully detailed. They ought to give attention to justifying their chosen methodology in terms of demonstrating applicability, adjustment, and usefulness in the paper. Likewise, the authors ought to refine the article by considering “scientific writing techniques” (i.e. building an argument; designing research questions; quotation rules, outlining sections and sub-sections, and so forth). Thus, I would encourage the authors to undertake some revisions, corrections, and paraphrasing works that may take some time. 

We thank the reviewer for reminding us of these general principles of methodology reporting in a scientific article. Nevertheless, we believe that we have met this objective with a very precise description of the methods used. 

 (6) All in all, I recommend the authors to reconsider the approach adopted here; think about the main empirical question they wish to examine; make sure the literature review is a lot more cohesive, and make sure the link between the research question and empirical results is a lot 'tighter' than presented herewith. 

We are not sure to understand what is being suggested here. 

(7) MDPI - Int. J. Environ. Res. Public Health is a remarkable peer-reviewed journal. A referee ought to recommend a manuscript that is deemed a great contribution to the journal’s future achievements. Consequently, the paper demonstrates rigorous research outcomes/findings that can be useful for the readers of the MDPI - Int. J. Environ. Res. Public Health. 

We thank the reviewer for this comment and more broadly for his valuable advices.